# Potential of a SAR Small-Satellite Constellation for Rapid Monitoring of Flood Extent

**Natsumi Kitajima [1,\*], Rie Seto [1], Dai Yamazaki [2], Xudong Zhou [2], Wenchao Ma [2]⊙ and Shinjiro Kanae [1]**

1  Department of Civil and Environmental Engineering, School of Environment and Society, Tokyo Institute of Technology, 2-12-1 Ookayama, Meguro-ku, Tokyo 152-8550, Japan; seto.r.ac@m.titech.ac.jp (R.S.); kanae@cv.titech.ac.jp (S.K.)
2  Institute of Industrial Science, The University of Tokyo, 4-6-1 Komaba, Meguro-ku, Tokyo 153-8505, Japan; yamadai@iis.u-tokyo.ac.jp (D.Y.); x.zhou@rainbow.iis.u-tokyo.ac.jp (X.Z.); wma@iis.u-tokyo.ac.jp (W.M.)
\*  Correspondence: kitajima.n.aa@m.titech.ac.jp

**Abstract:** Constellations of small satellites equipped with synthetic aperture radar (SAR) payloads can realize observations in short time intervals independently from daylight and weather conditions and this technology is now in the early stages of development. This tool would greatly contribute to rapid flood monitoring, which is usually one of the main missions in upcoming plans, but few studies have focused on this potential application and a required observation performance for flood disaster monitoring has been unclear. In this study, we propose an unprecedented method for investigating how flood extents would be temporally and spatially observed with a SAR small-satellite constellation and for evaluating that observation performance via an original index. The virtual experiments of flood monitoring with designed constellations were conducted using two case studies of flood events in Japan. Experimental results showed that a SAR small-satellite constellation with sun-synchronous orbit at 570 km altitude, 30-km swath, 15–30° incidence angle, and 20 satellites can achieve 87% acquisition of cumulative flood extent in total observations. There is a difference between the results of observation performance in two cases because of each flood's characteristics and a SAR satellite's observation system, which implies the necessity of individual assessments for various types of rivers.

**Keywords:** flood monitoring; synthetic aperture radar; small satellite; satellite constellation; disaster management

## 1. Introduction

Swift assessment of the location and extent of damage caused by disasters is vital to real-time situational awareness and effective measures in the response phase of disaster management [1,2]. In the case of flood disasters, adverse weather conditions and surface inundation typically obstruct field surveys, especially at night. Remote sensing technologies can offer useful solutions in such situations; in particular, satellite-based synthetic aperture radar (SAR) has great potential for flood detection [3]. Unlike optical sensors, SAR can penetrate clouds and rain and collect observations independently from sunlight [4]. Moreover, satellite as a platform facilitates observation over wide swathes and global coverage of remote areas, which cannot be achieved with aircrafts or unmanned aerial vehicles (UAVs) [5]. SAR satellites thus play an important role as powerful tools for flood detection across vast regions under unfavorable weather conditions by day or night.

Most SAR satellites operate in a sun-synchronous orbit (SSO), a type of near-polar orbit. The orbit's SSO geometry is kept nearly fixed with respect to the sun [6], so satellites in SSO are synchronized to consistently remain in the same position relative to the sun. That is, the satellite visits the same spot at almost the same time twice daily, for example, at 12:00/00:00 every day. Thanks to these features, SSO provides global coverage at all

latitudes (except for just a few degrees from the poles) [6], and enables constant observation at the same local time. Each SSO can be identified by its mean local time (MLT) [7].

However, the observation opportunities offered by current SAR satellite systems are insufficient from the perspective of rapid flood monitoring in the context of disaster management. Currently, most civilian SAR satellites operate in SSO with 06:00/18:00 MLT (the so-called dawn–dusk orbit) [8] due to its efficiency regarding solar power generation, whereas few other SAR satellites travel in SSO with other MLTs (e.g., ALOS-2 with 12:00/00:00 and NovaSAR with 10:30/22:30). Therefore, the observation opportunities currently provided by SAR satellite systems are biased, even though any location worldwide may be observed twice daily by each satellite. In the context of flood monitoring, the disaster management community requires flood extent information with little latency and frequent updating, but inadequate satellite revisit time is one issue preventing the collection of this information [5,9]. The dynamic flood process, which may both expand and shrink, is difficult to track within the limited time available.

The development of small satellites equipped with SAR could be key to resolving this issue. Small satellites have attracted considerable attention due to their wide-ranging applicability, supported by technological advancements in space engineering over the last decade [10]. These satellites cost significantly less than conventional large satellites, and this enables 'constellation' operation involving several satellites; by working in concert, groups of satellites have an excellent capacity for high revisit rates and short revisit time [10]. One example is Planet, which operates more than 200 optical small satellites. However, the miniaturization of SAR satellites has been slow compared to that of optical satellites due to the challenging design requirements of small SAR satellites, such as larger antennae and higher power throughputs [11]. In recent years, however, the miniaturization of electronic components and recent technological advances have finally ensured the compatibility of SAR with small platforms [12]. Constellations of SAR-equipped small satellites are in the early stages of development, and application-driven satellite design is favored in this early period [12]. In other words, each space mission can be optimized through the adoption of the appropriate constellation design and SAR device [13]. That is, the required observation performance (such as the revisit time, spatial coverage, swath, spatial resolution, and spectral bandwidth) should be specified in advance for a functional implementation.

Now several space companies such as QPS, Synspective, ICEYE, Capella Space, and Umbra Lab are planning to realize the constellation of SAR-equipped small satellites. Most of these plans considered flood monitoring as one of the main targets, but they did not discuss the required observation performance in the application of effective flood monitoring. Identification of this requirement needs an integrated approach that simulates the satellites orbiting in space and river flooding on the ground to investigate how flood extent would be temporally and spatially observed with a SAR small-satellite constellation. Therefore, in this study, we combined the design of constellations and the hourly outputs of a flood inundation simulation to conduct a virtual experiment of how a constellation of SAR small-satellites would observe the fluctuation of flood extents in spatial and temporal scales, using two flood events in Japan as case studies.

## 2. Methods

### 2.1. Design of SAR Small-Satellite Constellation

#### 2.1.1. Constellation Configuration

In this method, the Walker Constellation (WC) method was applied in the first step in designing SAR small-satellite constellations. The WC method consists of three integer parameters $T/P/F$, where $T$ denotes the total number of satellites, $P$ is the number of orbit planes, and $F$ is the relative phase difference between satellites in adjacent planes [14,15]. The details are described in Appendix A.

### 2.1.2. Orbital Elements

To determine the orbital elements of each satellite constituting a virtual constellation, we applied two-line element (TLE) sets of an existing SAR-equipped small satellite ICEYE-X2. The details are explained in Appendix B.

### 2.1.3. SAR Satellite Observation

SAR satellites observe the area of interest by adjusting the direction of their sensors. The incidence angle $\phi$ and the off-nadir angle $\psi$ are defined as shown in Figure 1. $\phi$ is typically larger than $\psi$ as a result of the curvature of the Earth. Supposing that the ground area that can be observed by that adjustment is an observable range $L$ [km] and assuming that the satellite orbit is circular and that the Earth's surface is spherical, $L$ is calculated as follows [16]:

$$L = R_e(\phi_{ub} - \psi_{ub}) - R_e(\phi_{lb} - \psi_{lb}) \tag{1}$$

where $R_e$ [km] denotes the radius of the Earth, $\phi_{ub}$ [rad] is the upper bound of incidence angles, $\psi_{ub}$ [rad] is the upper bound of off-nadir angles, $\phi_{lb}$ [rad] is the lower bound of incidence angles and $\psi_{lb}$ [rad] is the lower bound of off-nadir angles. A relationship between an incident angle and an off-nadir angle may be observed as follows [16]:

$$\frac{sin\phi}{sin\psi} = \frac{R_e}{R_e + a} \tag{2}$$

where $a$ [km] denotes the satellite's altitude. It is derived from Equations (1) and (2) that an observable range is getting wider as a satellite's altitude becomes higher.

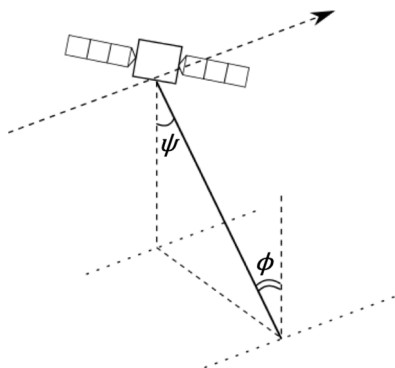

**Figure 1.** Schematic illustration of SAR satellite observation.

To define the incidence angle $\phi$ range and the satellite altitude $a$, the orbit and SAR property information of a SAR-equipped small satellite (ICEYE-X2) was used (detailed in Table 1) [17]. As shown in Table 1, an incidence angle from 15° to 30° may be used in its Stripmap imaging mode, and it operates at an altitude of 570 km, so we determined $\phi_{lb} = 15°$, $\phi_{ub} = 30°$, and $a = 570$ km for the calculation of $L$. Regarding the inclination, we supposed a general use of satellites including other applications in this paper, so a polar orbit was adopted, not a lower orbital inclination. Current SAR systems are capable of operating in different imaging modes by controlling the antenna radiation pattern, which practically results in different combinations of the swath width and resolution [18]. The most fundamental mode is the Stripmap mode, where the ground swath is illuminated with a continuous sequence of pulses while the antenna beam is fixed in its orientation, thus imaging a long strip parallel to the flight direction [18]. This imaging mode should be ideal in this analysis because it has 30 km swath and 3 m resolution properties that are suitable for the requirements for flood detection. For a better azimuth resolution, the Spotlight mode is utilized, but this operation is usually at the expense of spatial coverage (1-m resolution and $5 \times 5$ km$^2$ scene in the reference SAR-equipped small satellite). For a wider swath, the system can be operated in the ScanSAR mode, but the azimuth resolution is degraded

when compared to the Stripmap mode (this mode is currently under development in the reference SAR-equipped small satellite, but with 100 m resolution and 350 km swath in a traditional SAR satellite ALOS-2) [17].

**Table 1.** Information of the reference SAR small-satellite.

| Swath | 30 km$^2$ |
|---|---|
| Look direction | both LEFT and RIGHT |
| Incidence angle range | 15.0°–30.0° |
| Nominal altitude | 570 km |
| Inclination | 97.69° |
| Ground track repeat | 18 days |

*2.2. Assessment of Revisit Rates and Revisit Time*

As a preliminary step for the subsequent virtual experiment in flood monitoring using the designed SAR small-satellite constellation, we sought the appropriate $T/P/F$ parameter settings in the WC method. Although the potential combinations of these parameters are finite (assuming that $F$ is an integer number), predetermination of the parameter setting that is likely to perform well for each $T$ is advisable to ensure effective analysis. Thus, revisit rates and revisit time were assessed for all combinations of $T/P/F$ for each $T$. Here, we provisionally tried to determine the settings of $T$ that were likely to accomplish the revisit time in the order of one to several hours. To investigate the area to be observed by each SAR small-satellite constellation, the mapping of observable range $L$ along the satellites' passes was simulated for the period of the recurrent days—18 days in this case—as listed in Table 1. Then, several representative points were prepared to check the observation timings. We named these points ground check points (GCPs) and set GCPs at five locations in Japan: Sapporo, Sendai, Tokyo, Osaka, and Fukuoka, whose locations are depicted in Figure 2 (left). Figure 2 (right) presents an example of the ground tracks of satellites (purple line), observable range $L$ (yellow belt zone), and swath width (orange belt zone) both in an ascending pass and a descending pass. The revisit rates and revisit time can be calculated by extracting the date and time when the observable zone catches the GCPs. Because the calculation period was set at 18 days, the daily revisit rates at each GCP was calculated by dividing the total number of observations for that point by 18. For the revisit time, the mean revisit time was calculated by averaging the time interval between observations for all GCPs and the maximum revisit time was calculated by extracting the largest time interval between observations for all GCPs.

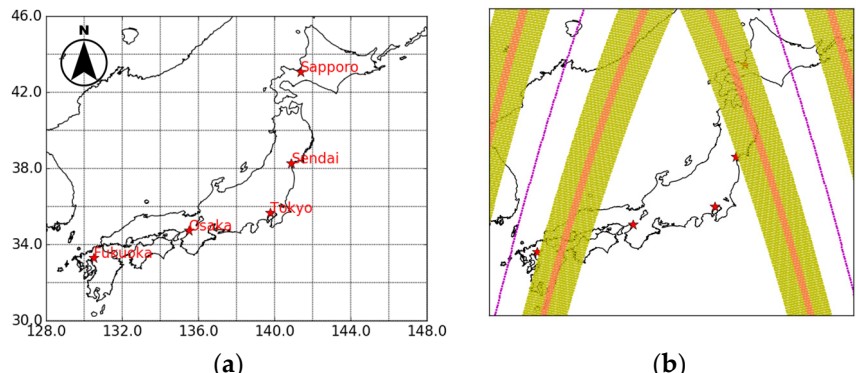

(**a**) (**b**)

**Figure 2.** (**a**) The locations of 5 GCPs; (**b**) the ground tracks of a satellite (purple line), observable range L (yellow belt zone), and swath width (orange belt zone) in an ascending pass and a descending pass.

*2.3. Virtual Experiment of Flood Monitoring*

2.3.1. Time Series Data of Flood Extent

We investigated how a time series fluctuation of flood extent could be observed by the virtual SAR small-satellite constellation. However, as mentioned above, observation

data with a high temporal resolution are currently unavailable. We therefore prepared the numerical simulation data of hourly flood depth from the catchment-based macro-scale-floodplain (CaMa-Flood) model, and assumed that these simulation outputs can be used as virtual truth of flood fluctuation in this analysis. The CaMa-Flood model is designed to simulate hydrodynamics in continental-scale rivers [19]. CaMa-Flood routes input runoff generated by a hydrological or land surface model to oceans or inland seas along a prescribed river network map. It calculates river and floodplain water storage, river discharge, water depth, and inundated area at each grid point. The flow characteristics are calculated with the local inertial equations along the river network map. The detailed model structures and parameters can refer to the original paper of Yamazaki et al. [19]. The hourly runoff data were input from Today's Earth Japan, with a land surface model MATSIRO and forcing inputs from MSM-GPV [20,21]. This simulation does not take into account anthropogenic water management, such as dam operation and reservoir effect, but this can be considered irrelevant to this virtual experiment; the simulation outputs still provide sufficient reference for the likely behavior of flooding observed with satellites.

In the flood simulation, the spatial resolution was downscaled to 30 m. As noted in the previous section, the designed SAR small-satellite constellation is assumed to use Stripmap mode with a resolution of 3 m (and a corresponding observation width of 30 km). Although this resolution does not coincide with those of simulation outputs, this may not be a problem because the imagery resolution represents a size per pixel rather than the discriminable size. For a specific object to be recognized, the object must typically be 10–20 times larger than the resolution. Taking this point into account, even if SAR data were actually acquired with 3 m resolution, the flood extent that emerges after image processing would cover an area of at least $30 \times 30$ m$^2$. Therefore, the resolution of simulation outputs may be considered reasonable for this analysis.

### 2.3.2. Operation Rule of Satellites

In this virtual experiment wherein the designed SAR small-satellite constellation was operated to monitor flood fluctuation, we investigated the maximum potential of that observation. Due to the various satellite locations and the finite swath width—30 km in this case, based on the properties of the reference small satellite—in some instances, the entire flood extent area could not be observed in this analysis. Therefore, we adjusted the observation angle from 15° to 30° and between left- and right-looking at each observation time so that each satellite could orient its SAR instrument toward a target region and observe the largest possible extent of flood area from its location.

Concerning the limitations of the observations of SAR-equipped small satellites, we make the following two assumptions. First, disaster observation is prioritized, given that even satellites owned by private companies are typically used in emergencies. Second, concerning limitations of the power system, energy generation must be equal to or greater than the energy consumption of the SAR imaging mission sequence [22]. On the basis of these assumptions and the assumption that the interval between observations at daytime and night is about 12 h, small satellites can store sufficient power if disaster management is prioritized.

### 2.3.3. Case Studies of Flood Events in Japan

One flood event included in the case study was heavy rainfall in the Kyushu region in July 2012. The torrential rainfall associated with the rain front caused floods and landslides in the northern Kyushu region, mainly in Kumamoto, Oita, Fukuoka, and Saga prefectures from 11–14 July 2012. There were 5 points where the total amount of rainfall during 4 days was more than 500 mm, such as 816.5 mm at the Aso city of Kumamoto, 616.5 mm at the Hita city of Ohita, 649.0 mm at the Yame city of Fukuoka. The rainfall event caused 32 fatalities and damaged 13,263 houses (769 collapsed and 12,494 were inundated). In this case study, the Chikugo River basin (33° N–34° N/130° E–131° E) was selected for analysis.

The other flood event included was Typhoon No. 19 in October 2019, which caused heavy rain, stormy winds, high waves, and storm surges across an extensive area from 10–13 October 2019. There were 17 points where the total amount of rainfall was more than 500 mm, such as 1001.5 mm at the Hakone town of Kanagawa, 760.0 mm at the Izu city of Shizuoka, 687.0 mm at the Chichibu city of Saitama. River flooding and landslides resulted in 99 fatalities, and 4008 houses completely or partially collapsed while 70,341 houses were inundated. In this case study, we focused on the Chikuma River basin (36° N–37° N/138° E–139° E), where 5086 houses were inundated.

## 3. Results

### 3.1. Assessment of Revisit Rate and Revisit Time

The revisit rate, mean revisit time, and maximum revisit time were calculated from each result of the five GCPs. Based on this calculation, we found that the mean revisit time achieved was around 7 h with $T = 8$, 5 h with $T = 12$, 3 h with $T = 20$, 2.5 h with $T = 24$, 2 h with $T = 28$, 1.5 h with $T = 40$, and 1 h with $T = 56$. These results suggest a nearly linear correlation between the number of satellites and the revisit time. We speculated that this correlation has its roots in the constellation design process, wherein the RAAN and MA were arranged symmetrically at even intervals.

To assess the difference between the parameter settings of $P$ and $F$, the revisit rates, the mean revisit time, and the maximum revisit time as a function of $P$ for each number of $T$ were set as shown in Figure 3. Here, the optimal parameter setting of $F$ was set for each $P$, so the figure allows us to evaluate the parameter settings of $T/P/F$ that yield the best performance in terms of revisit rates, mean revisit time, and maximum revisit time when the number of satellites ($T$) is given. As these results indicate, the revisit rates and mean revisit time remain almost unchanged with the variation of $P$, while the maximum revisit time exhibits different behavior that roughly improves as $P$ increases. Based on these results, Table 2 lists the best parameter settings of $P$ and $F$ for the revisit rates, the mean revisit time, and the maximum revisit time, which are for each $T$ the largest one in Figure 3a, the shortest one in Figure 3b, and the shortest one in Figure 3c, respectively. As shown in Table 2, for revisit rates and mean revisit time, the product of these two values nearly equals to 24 h. It was also found that the optimal settings of $P$ and $F$ are mostly the same between these two values with an exception of $T = 24$. This point should be more deeply examined in the further analysis with a higher number of GCPs in broad areas.

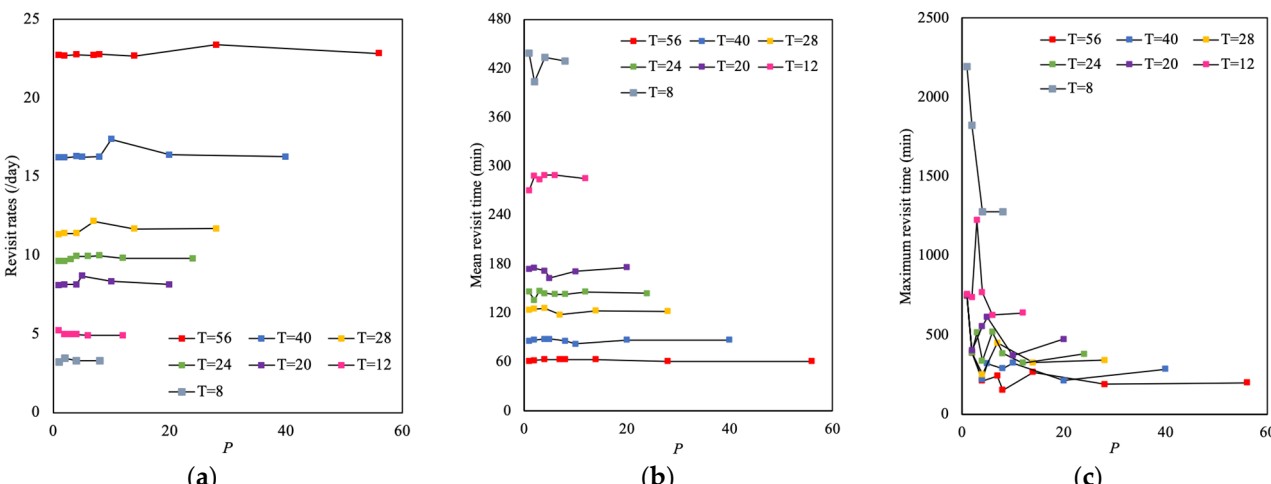

**Figure 3.** Results as a function of $P$ for: (**a**) Revisit rates; (**b**) Mean revisit time; (**c**) Maximum revisit time.

**Table 2.** The best parameter setting of *P* and *F*, and the corresponding results of the mean revisit rates, mean revisit time, and maximum revisit time for each *T*.

| *T* | *P* | *(F)* | Revisit Rates (/day) | *P* | *(F)* | Mean Revisit Time | *P* | *(F)* | Maximum Revisit Time |
|-----|-----|-------|----------------------|-----|-------|-------------------|-----|-------|----------------------|
| 8   | 2   | (1)   | 3.47                 | 2   | (1)   | 6 h 44 min        | 4   | (2)   | 21 h 16 min          |
| 12  | 1   | (-)   | 5.22                 | 1   | (-)   | 4 h 30 min        | 6   | (2)   | 10 h 25 min          |
| 20  | 5   | (4)   | 8.68                 | 5   | (4)   | 2 h 43 min        | 10  | (7)   | 6 h 11 min           |
| 24  | 8   | (7)   | 9.98                 | 2   | (1)   | 2 h 16 min        | 12  | (7)   | 5 h 23 min           |
| 28  | 7   | (2)   | 12.16                | 7   | (2)   | 1 h 58 min        | 4   | (3)   | 4 h 11 min           |
| 40  | 10  | (9)   | 17.37                | 10  | (9)   | 1 h 22 min        | 20  | (1)   | 3 h 32 min           |
| 56  | 28  | (10)  | 23.38                | 28  | (10)  | 1 h 1 min         | 8   | (1)   | 2 h 32 min           |

Based on the above results, we attempted to determine the parameter settings to be used in the subsequent virtual experiment. In terms of emergency observation during floods, the maximum revisit time is a critical factor in avoiding missing an observation timing as soon as possible. Therefore, we decided to adopt the optimal parameter setting for the maximum revisit time. However, a single setting for each *T* may not be sufficient for the evaluation of the virtual experiment, so we also took the optimal parameter setting for the revisit rates, because this was calculated using a different method than that used for the former.

*3.2. Virtual Experiment of Flood Monitoring*

The virtual experiment of flood monitoring using a SAR small-satellite constellation was executed in the two flood event case studies. Because flood simulation outputs also included the permanent water area in the river channel, we masked this area in advance. Figure 4 presents the results of the virtual observation of flood extent with each constellation (*T* = 8, 12, 20, 24, 28, 40, and 56) in case 1 (heavy rainfall in the Kyushu region from 14:00 on July 12 to 14:00 on 13 July 2012). In this figure, the simulated flood extent at the time of observation is represented by a blue line, and the observed simulated flood extent is represented in a multi-colored bar graph. This multi-color graph expresses the proportion of flood extent that was observed for the first time (1st), second time (2nd), third time (3rd), fourth time (4th), fifth time (5th), and sixth time and above (>6 th) after the flood occurred at 14:00 on 12 July 2012. In other words, these components express how much new information (1st) and overlapped information (2nd to >6th) regarding flood extent are acquired at each observation with respect to the past observations.

Here, we also introduced the damage extent and the observed damage extent. Damage extent was defined as the area that had experienced flooding up to each time in this event, so it was assumed to have already been damaged. In other words, the damage extent is the cumulative flood extent, including the area where flood water has already receded. Observed damage extent denotes how much damage extent area was acquired as a result of multiple observations until each time. This is calculated as the summation of the '1st' first area until each time because it gives the cumulative observed flood extent with no redundancy. Comparison of the two dotted lines of damage extent and observed damage extent demonstrates the extent to which the observation system can trace the real situation on a spatial and temporal scale. Figure 5 presents the results for case 2 (Typhoon No. 19 from 13:00 on 12 October 2019 to 11:00 on 13 October 2019). Here, the definition of legends is the same as that in Figure 4.

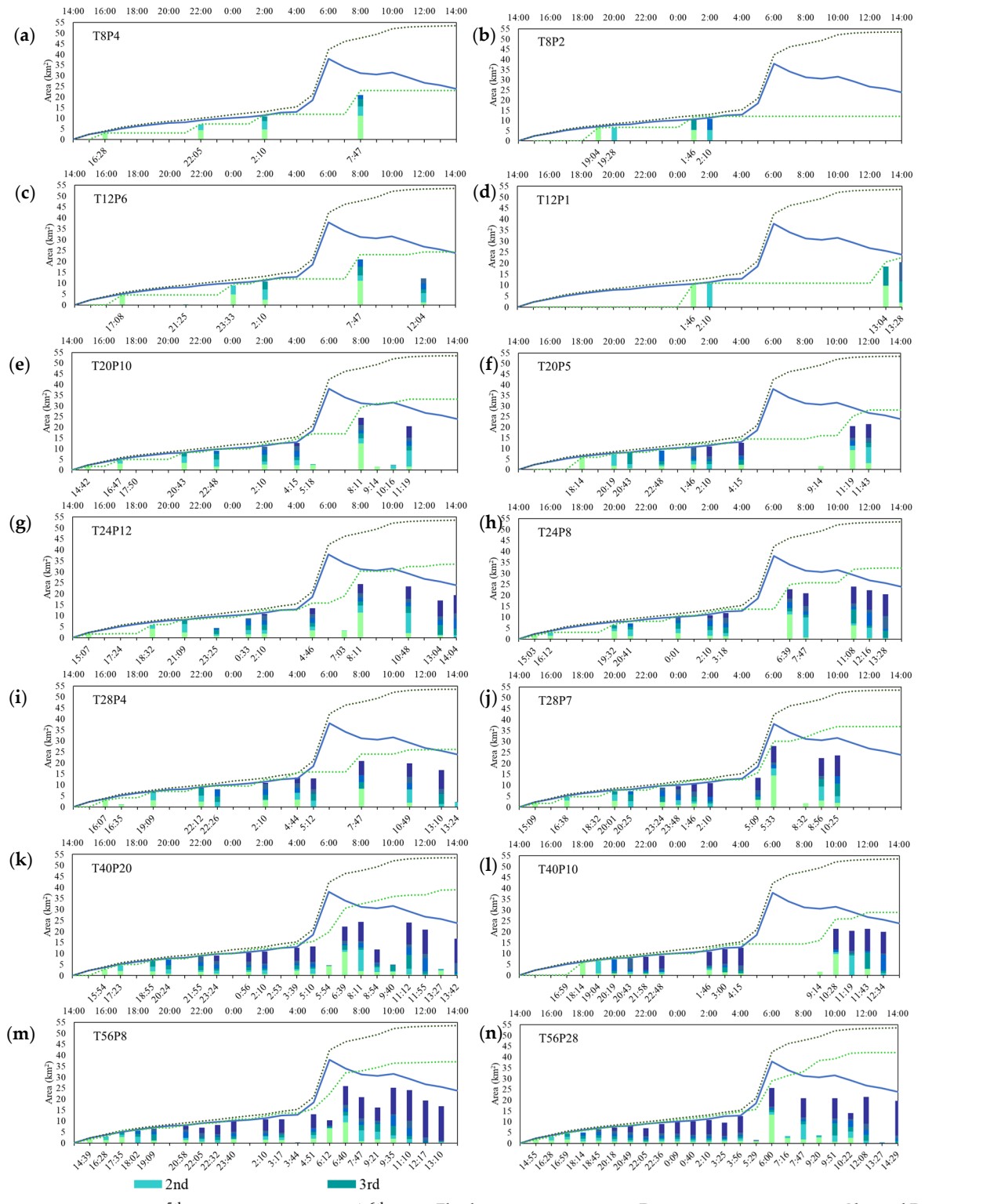

**Figure 4.** The virtual observation of the simulated flood extent with constellations of (**a**) *T* = 8, *P* = 4; (**b**) *T* = 8, *P* = 2; (**c**) *T* = 12, *P* = 6; (**d**) *T* = 12, *P* = 1; (**e**) *T* = 20, *P* = 10; (**f**) *T* = 20, *P* = 5; (**g**) *T* = 24, *P* = 12; (**h**) *T* = 24, *P* = 8; (**i**) *T* = 28, *P* = 4; (**j**) *T* = 28, *P* = 7; (**k**) *T* = 40, *P* = 20; (**l**) *T* = 40, *P* = 10; (**m**) *T* = 56, *P* = 8; (**n**) *T* = 56, *P* = 28 in case 1 (heavy rainfall in the Kyushu region from 14:00 on 12 July 2012 to 14:00 on 13 July 2012).

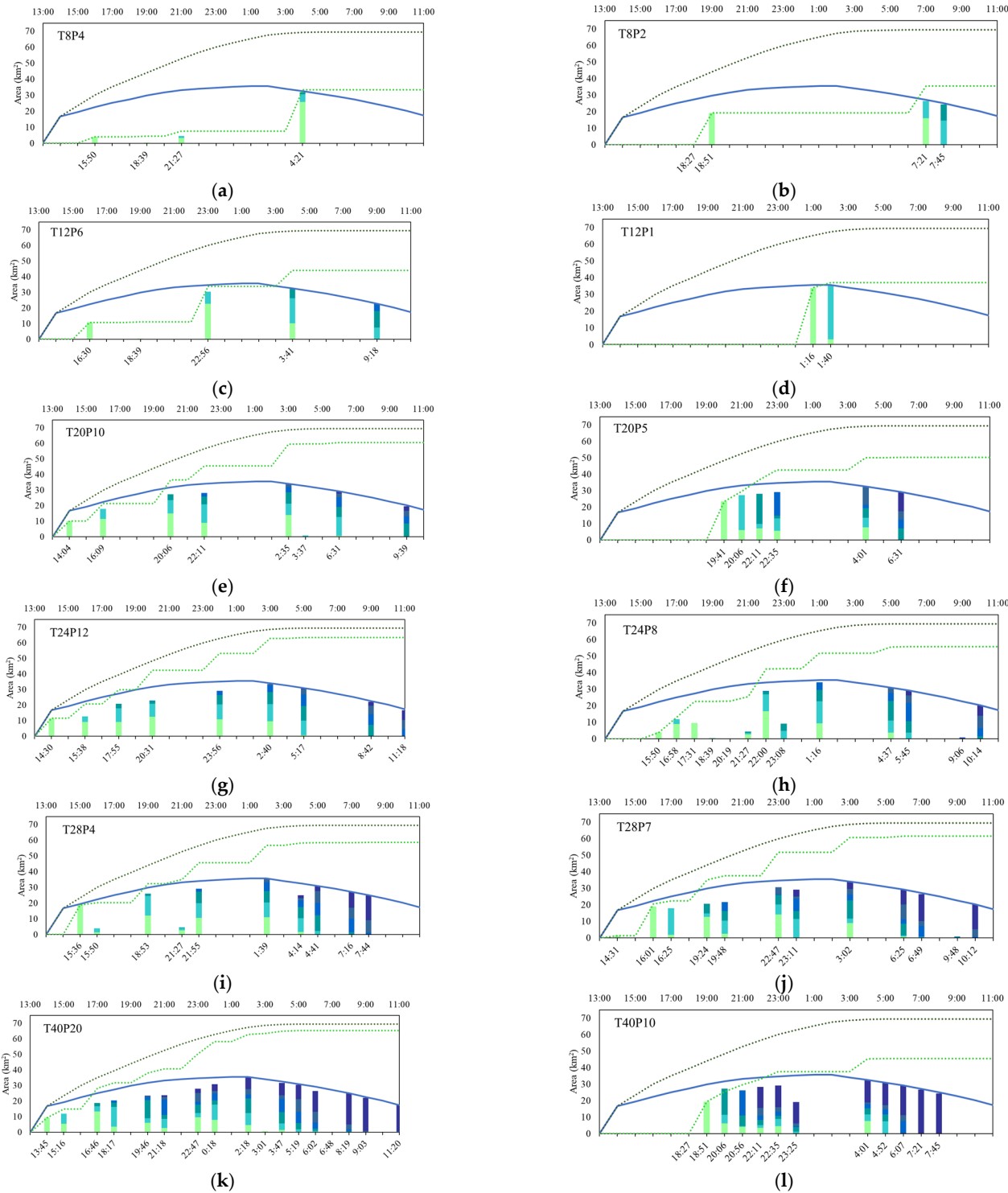

**Figure 5.** *Cont.*

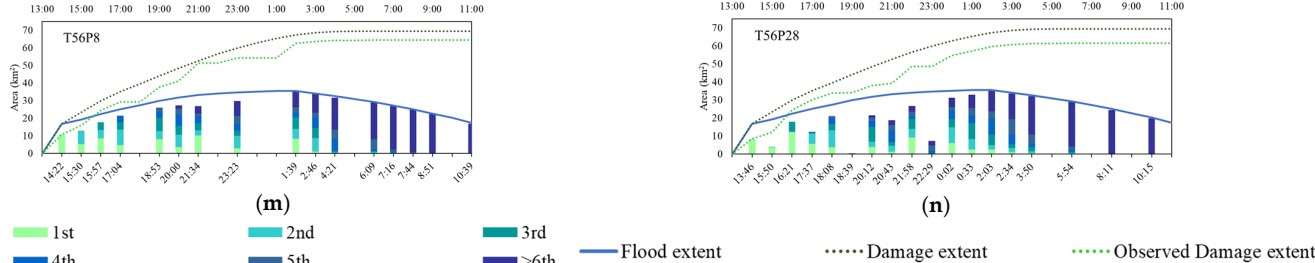

**Figure 5.** The virtual observation of the simulated flood extent with constellations of (**a**) *T* = 8, *P* = 4; (**b**) *T* = 8, *P* = 2; (**c**) *T* = 12, *P* = 6; (**d**) *T* = 12, *P* = 1; (**e**) *T* = 20, *P* = 10; (**f**) *T* = 20, *P* = 5; (**g**) *T* = 24, *P* = 12; (**h**) *T* = 24, *P* = 8; (**i**) *T* = 28, *P* = 4; (**j**) *T* = 28, *P* = 7; (**k**) *T* = 40, *P* = 20; (**l**) *T* = 40, *P* = 10; (**m**) *T* = 56, *P* = 8; (**n**) *T* = 56, *P* = 28 in case 2 (Typhoon No. 19 from 13:00 on October 12 to 11:00 on 13 October 2019).

### 3.3. Evaluation of Observation Performance

Specific observation behavior offered by each constellation was already shown in Figures 4 and 5, but it is still difficult to judge whether each observation performance is sufficient or not in the context of flood disaster management. In order to evaluate this observation performance quantitatively in terms of the damage assessment in flood disasters, an original performance index was devised in this paper. This performance index was defined as the ratio of observed damage extent to damage extent in the final phase, and it was named the 'capture ratio.' The capture ratio describes how much area of damaged extent was acquired as a result of time series observation excluding overlapped information. The main reason why we defined this value as the performance index is that the capture ratio has its highest limit, which is 1 (= 100%). This limitation is crucial from the perspective of cost estimation in the practical implementation of SAR small-satellite constellation. The observation performance is generally improved as the number of satellites increases, but the larger number of satellites requires a more expensive cost. Therefore, it is difficult to determine the optimal condition of the constellation if the performance index is getting higher infinitely. For example, the total area of observed flood extent has no limitation, so it is not an appropriate indicator for the performance evaluation. On the other hand, the capture ratio (= observed damage extent/damage extent) must eventually reach its highest value. This is because observed damage extent consists of only the 1st area, and the accumulation of the 2nd > 6th areas does not contribute to this flood damage assessment anymore. In short, the capture ratio is a performance index that simultaneously expresses the sufficiency and efficiency of observation and it provides the comprehensive result of flood damage assessment on a spatiotemporal scale.

The calculation result of the capture ratio in case 1 and case 2 is presented in Table 3. Here, the result of one large satellite (*T* = 1) is also shown beside that of *T* = 8, 12, 20, 24, 28, 40, and 56. As can be seen from this table, the capture ratio usually grew higher as the number of *T* increased, but it varied with the number of *P* even under the same number of *T*, and sometimes the smaller number of *T* has a better result for the capture ratio. This implies that the number of satellites (= *T*) is of course a dominant parameter that determines observation performance, but the configuration of satellite constellation (= *T*/*P*/*F*) should be also carefully selected. This performance index, the capture ratio, allows us to compare the result of multiple case studies. When we focus on the difference between case 1 and case 2, it can be seen that in case 1 more than 56 satellites are required to achieve 81% acquisition, while even 20 satellites accomplish 87% in case 2. This is considered to be mainly because of the characteristics of each flood event. In case 1, the peak flooding was happening in a short time, which means flood extent was expanding and shrinking quickly. Thus, if the observation timing was missed or the observed area was insufficient in that peak time, data acquisition would be difficult to recover. On the other hand, in case 2, the flood was gradually expanding and shrinking, so it was easier to capture most of the damage extent during that period. This result demonstrated that

observation performance is fairly dependent on temporal and spatial flood fluctuation in each event. As another interesting point, in fact, the whole area of damage extent was larger in case 2 (69.49 km$^2$) than that in case 1 (53.82 km$^2$), but the capture ratio was generally higher in case 2, so it seems unnatural. This is because there is a water channel network unique to the region (Saga city) in case 1. Due to that geographical feature, the flood extent was distributed more widely, but the total area resulted in a smaller value, which means it was difficult to capture the whole area of flood extent at one time (but conversely it provided more opportunities to observe a part of the flood extent). This kind of flood feature also impacted on the observation performance.

**Table 3.** The performance index of the capture ratio in case 1 and 2.

|  | T1 (Large Sat.) | T8 P4 | T8 P2 | T12 P6 | T12 P1 | T20 P10 | T20 P5 | T24 P12 | T24 P8 | T28 P4 | T28 P7 | T40 P20 | T40 P10 | T56 P8 | T56 P28 |
|---|---|---|---|---|---|---|---|---|---|---|---|---|---|---|---|
| Case 1 | 0.00 | 0.50 | 0.39 | 0.55 | 0.42 | 0.62 | 0.54 | 0.63 | 0.61 | 0.62 | 0.70 | 0.78 | 0.55 | 0.76 | 0.81 |
| Case 2 | 0.16 | 0.48 | 0.51 | 0.63 | 0.54 | 0.87 | 0.74 | 0.91 | 0.80 | 0.85 | 0.89 | 0.94 | 0.75 | 0.96 | 0.92 |

Therefore, the required observation performance would vary depending on how dynamically and widely each flood process would occur on a spatial and temporal scale, which would be determined by meteorological conditions and topographical features.

## 4. Discussion

In both cases 1 and 2, the observed flood extent in some instances was insufficient relative to the flood extent at each time. To investigate this issue, Figure 6 presents the captured flood extent with the constellation of $T = 56$, $P = 28$ in case 1. The red zone represents the observed extent, whose swath width is 30 km. Here, it is clear that the observed area did not always match the flood area—for example, at 5:29, 7:16, and 9:20—and this discrepancy caused insufficient data acquisition. This is generally attributable to two factors. The first factor is the suitability of satellite passes: satellites do not always pass over an appropriate location with respect to each area of interest, so they are sometimes unable to capture it well. This is related to the second factor, which is the limited range of SAR's incidence angles. The look direction of a SAR instrument cannot be adjusted in all ranges, and this restriction prevents flexible operation according to the target area.

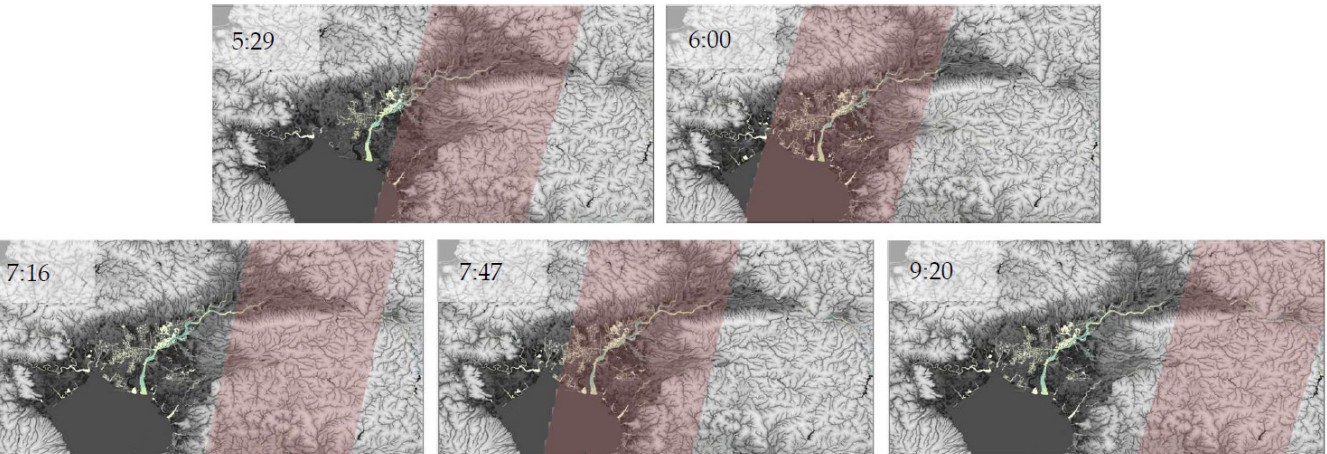

**Figure 6.** Observed flood area (the matched area of the red zone and the yellow–green–blue area) at 5:29, 6:00, 7:16, 7:47, and 9:20 on 13 July 2012 in the case of $T = 56$, $P = 28$.

However, some discrepancy seems inevitable, particularly for satellite observations collected using onboard SAR. Satellite passes are a predefined condition in their orbit and cannot be easily changed according to the area of interest. About the incidence angle, in

addition to its physical restriction, the range of angles is also bounded by the accuracy of flood detection [23]. That is, an incidence angle that is too large or too small would disturb the interpretation of SAR images, resulting in reduced flood detection accuracy. Therefore, the beam angle should be maintained within a specific range to guarantee SAR observation data quality.

One possible way to mitigate the technical challenges of SAR systems would be to expand the swath width. Currently, small satellites can only be equipped with antennae that are smaller than those used for large satellites, making it more difficult to achieve wider swaths. However, technological innovations are expected to overcome this issue: small satellites can go through more development cycles than larger, traditional satellites, and they present more opportunities for harnessing new technologies [12]. Another way to mitigate the technical challenges should be implemented in the operational phase, regardless of technological innovations. As shown in Figures 4 and 5, even low levels of acquisition can sometimes contribute to improving the extent of observed damage—for instance, at 5:54 in T40P20 of case 1 and at 17:31 in T24P8 of case 2. This implies that a suitable location of the observation target area relative to both the present situation and past observations is critical for better situational awareness, even if it is bounded by the restrictions mentioned above. Furthermore, this suggests that careful selection of the target area should be implemented by taking into account possible future flood situations, because the flood extent at each time does not appear prior to observation. In this regard, flood forecasting systems based on other information, such as precipitation, river discharge, and water level, should become useful tools for predicting which areas are likely to be affected and allowing proper operational commands to be sent to each satellite in advance. In the case of multi-location flood events, the required operation for a selection of target areas might be more complicated. Thus, an integrated operation system combined with flood forecast is desirable for effective observation by means of SAR small-satellite constellations. At the same time, the collection of flood extent data in high spatiotemporal resolution might also be useful in improving flood simulation and forecasting in turn. To realize such an observation system in the future, more interactive cooperation and collaboration between the satellite, the hydrology and the disaster management field should be essential.

## 5. Conclusions

In this study, we investigated how flood extents could be observed using SAR small-satellite constellation on a spatial and temporal scale. In the first step, we designed constellations using the Walker Constellation method, with parameters of $T$ (total number of satellites), $P$ (number of equally spaced orbit planes), and $F$ (relative phase difference between satellites in adjacent planes). Then, the designed SAR small-satellite constellations were applied in a virtual flood monitoring experiment involving numerical simulations of two case studies of flood events in Japan.

Each constellation's observation behavior was successfully presented, and an original performance index was introduced to evaluate the flood monitoring observation performance. The results demonstrated that a SAR small-satellite constellation with sun-synchronous orbit at 570 km altitude, 30 km swath, 15–30° incidence angle, and 20 satellites can achieve 87% acquisition of the damage extent (cumulative flood extent) in a time series observation in one case. Comparing the results of two cases, it was found that the observation performance depends on each flood's characteristics and it is related to some features of the SAR small-satellite observation system.

The results of two case studies in Japan under a condition with a specific satellite and orbit, which are shown in this paper, would be insufficient to conclude definitively the specific requirements of a SAR small-satellite constellation in terms of flood disaster monitoring. However, it was suggested that an individual assessment would be needed for each flood case in various regions with several kinds of satellite and orbit (e.g., a lower orbital inclination which is focusing on low and middle latitudes), since the optimal parameter settings are different in each case. In this regard, our proposed approach can be

applied to any flood cases in regions all over the world, so this study should be useful in that it built the first framework and established the evaluation method for future analysis. Therefore, based on this research, more comprehensive analyses in other areas with other small SAR satellites should be performed to determine the universal applications as needed.

**Author Contributions:** Conceptualization and methodology, N.K., R.S., and S.K.; software and formal analysis, N.K.; resources, D.Y., X.Z., and W.M.; writing—original draft preparation, N.K.; writing—review and editing, R.S. and S.K.; supervision, S.K. All authors have read and agreed to the published version of the manuscript.

**Funding:** This research was supported by MEXT/JSPS KAKENHI 16H06291.

**Acknowledgments:** Today's Earth research products used in this paper were supplied by Japan Aerospace Exploration Agency and Institute of Industrial Science, The University of Tokyo.

**Conflicts of Interest:** The authors declare no conflict of interest.

## Appendix A

Appendix A describes the details of the Walker Constellation (WC) method and how it was implemented in our method. This is the most frequently used method in practice, and it enables continuous global coverage [14,15]. Among the several possible ways of designing constellation configurations such as Street of Coverage and Flower Constellation, the WC method simultaneously offers simplicity and functionality, which is appropriate for this first attempt. It is also suitable for our objective of constructing a virtual constellation based on the orbital elements of an existing satellite. The WC method is based on the assumption that all orbits are circular and that they have a common altitude and inclination with reference to the equator [14,15]. The right ascension of the ascending node (RAAN)—the place where the satellite crosses the equator from south to north—of each orbit plane is equally spaced on the equator plane, and all satellites are equally spaced on each orbit plane. This arrangement is defined with integer parameters $T$, $P$, and $F$, where $T$ denotes the total number of satellites, $P$ is the number of orbit planes and $F$ is the relative phase difference between satellites in adjacent planes. $P$ orbit planes are positioned at $\alpha = 180°/P$ intervals and each has $S = T/P$ satellites, which are placed at $\beta = 360°/S$ intervals on each orbit plane, given that $S$ is the number of satellites per plane. When all orbit planes have the same relative phase difference, $F$ is defined as an integer number to keep the arrangement's symmetry, and the angle difference between satellites in adjacent planes is $\gamma = F \times 360°/T$. For reference, Figure A1 (left) presents an example of a WC whose parameters are $T = 8$, $P = 2$, and $F = 1$, where $S = 8/2 = 4$, $\alpha = 180°/2 = 90°$, $\beta = 360°/4 = 90°$, and $\gamma = 1 \times 360°/8 = 45°$ with an inclination 90°.

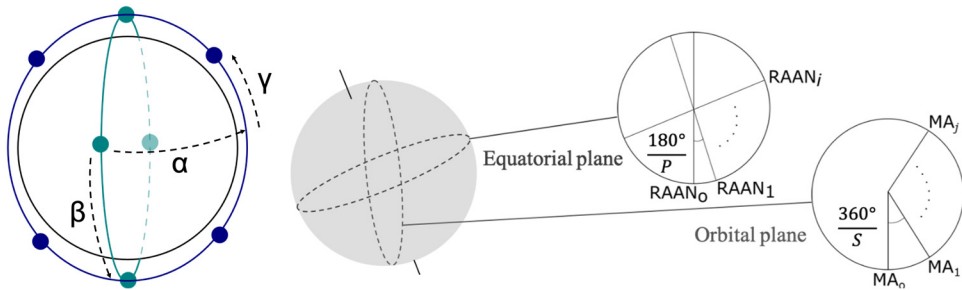

**Figure A1.** Schematic illustration of the Walker Constellation: (**left**) an example of whose parameters are $T = 8$, $P = 2$, and $F = 1$ with an inclination 90°. (**right**) Arrangement of RAAN and MA.

## Appendix B

Appendix B describes how the orbital elements of each satellite were defined in our method. We applied two-line element (TLE) sets, which are data formats comprising listed orbital elements. Because a TLE specifies an orbit's size and shape and how the orbit is

oriented with respect to the Earth, we can simulate each satellite's movement and compute its location at a specific time. For existing satellites, public information may be available, but for a virtual constellation that includes non-existent satellites, it is necessary to first define their TLE. To overcome this issue, we selected an existing satellite and adjusted some aspects of its TLE in accordance with the structure of the WC method. Here, ICEYE-X2—one of the SAR-equipped small satellites—was selected as a reference for TLE. This satellite offers global coverage as it is in SSO. It orbits the Earth 15 times per day, and the inclination of its orbit is 97.69° with reference to the equator [17].

Supposing the reference satellite's RAAN is $RAAN_o$ [°] and its mean anomaly (MA)—the position of satellites on each orbit plane—is $MA_o$ [°]. Following the WC method, as shown in Figure A1 (right), $RAAN_i$ [°]—the RAAN of the *i*-th orbit—and $MA_{i,j}$ [°]—the MA of the *j*-th satellite on the *i*-th orbit—can be calculated as follows [24]:

$$RAAN_i = RAAN_o + i\frac{180}{P} \tag{A1}$$

$$MA_{i,j} = MA_o + iF' + j\frac{360}{S} \tag{A2}$$

With regard to the other orbital elements in TLE, values identical to those of the reference satellite were used for this computation.

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
