# Peer review of "Potential of a SAR Small-Satellite Constellation for Rapid Monitoring of Flood Extent"

_remotesensing, doi:10.3390/rs13101959_

Round 1
Reviewer 1 Report
Improving temporal resolution by small SAR satellites constellation is helpful for disaster monitoring. In this communication, the authors show a constellation concept for flood extent monitoring. However, I think this article lacks of rigorous analysis and innovation. Some suggestions are shown below for consideration.
1) The selection of orbits is very important for obtaining valid data. However, the authors just use the parameters of ICEYE for structuring constellation without any application requirements analysis, and I think it is unconvincing. I suggest that the authors should take into account the optimal range of incident angles, swath width, revisiting interval, resolution, and sensitivity requirements of flood extent monitoring, and comprehensively consider the choice of orbit. Application demand is always the driving force of small satellite design.
2) There are too many definitions of basic concepts, which make this paper more like a science report or a scientific popular article than an academic paper, especially communication or letter.
3) The authors should consider more small satellite programs, such as Capella Space, Umbra Lab, and Synspective. Because Synspective is also a Japanese company, I think you can get more cooperation opportunities with it.
4) All the figures in this communication are not clear for reading.
5) As a communication, the length of this paper needs to be compressed.
Reviewer 2 Report
This manuscript presents a novel study on design of SAR small-satellite constellation and the performance evaluation for flood monitoring. The developed methods of both designing small-satellite constellation and evaluating the observation performance of a SAR constellation in flooding monitoring have instructive meaning for other applications and studies. The descriptions about the methods and experiment are concise and clear. But there are some points needing to be clarified.
Specific comments are:
- Table 1, the parameter F (Line 105) has not been clearly defined and explained. Current explanation about F, “Relative phase difference between satellites in adjacent planes”, is not well understandable. Why should it be an integer? Maybe an extra figure is needed to illustrate F.
- Table 3 is a little confusing. Firstly, it seems from Table 3 that there is a fixed relationship between “mean revisit rates” and “mean revisit time”, i.e. the product of the two values nearly equals to 24 hours. Please confirm this guess. Secondly, it seems that the optimal value of “mean revisit rates” and “mean revisit time” are reached synchronously under the same set of (P,F) in nearly all occasions when T is given. But there is one exception, i.e., T is 24, corresponding the fourth row of Table 3. An explanation is required here.
- Figure 4-8 are blurred, and higher resolution images are needed to replace them.
- There are some typewriting errors. For example, Line 368, “off course” should be “of course”. Line 104, “and have S = T/P satellites ” should be “ and each has S = T/P satellites ”. Please go through the main text and correct such kind of errors.
Reviewer 3 Report
This paper presents investigation flood phenomena with using synthetic aperture radar SAR small- satellite constellation on two examples of events in Japan. The paper is very interesting and presenting methods has potential to short application in flood forecasting. Introduction is well prepared and presents good motivations to perform of this study. I see strong novelty of this research, mainly in hydrological applications. I recommend this paper to accept after minor corrections:
1.I suggest removing table 1 because inside is not presented any important information beside explanations of parameters. I suggest these informations to add in to fig 1 as legend
- L130: Should be fig 1? or lack fig 2 in paper
- Chapter 2.3. please add more informations about CaMa-Flood model (input parameters, input parameters, how runoff and flood wahe propagations is calculated. These informations can be add to appendix.
- Chapter 2.3.2.As I understand, simulation of flood was performed for real precipitation? If yeas please add more informations about characteristics of rainfall events, sum of precipitation, intensity, frequency if it is possible
5.Discussion: it is difficult to say “observed flood event” because it is results of work of model . May be better is simulated flood events? In this case please also consider rewrite title of fig 6 and 7 where are “virtual observation”
Reviewer 4 Report
Dear Mr. Nikola Stojanovic,
Thank you for giving me the opportunity to review this MS. There are no issues of conflicting interest, and I have no personal or professional affiliation with the authors.
This manuscript presents a study on Potential of SAR Small-Satellite Constellation for Rapid Monitoring of Flood Extent. It is clear that the authors put a great deal of work into getting this manuscript to its present form. However, the English text it needs to be revised with the help of an English native speaker. The figures need extensive revision. Therefore, I suggest that the paper can be accepted after major revision. I appreciate to take these comments into account and please revise the comments in the attached file.
The following revisions are suggested:
Abstract:
- Line 21; “…altitude, a 30-km swath, a 15–30° incidence…”, delete the underlined.
- Line 20, please modify "an" to ""a" before "….. SAR small-satellite constellation"
Methods:
- The subsection “Deign of the SAR small-satellite constellation”; there is no need to include all these details about design, especially equations from 1 to 4, I cannot see their relation to the results. However, if you used them to get results, please clarify this.
- Line 92, please add reference.
- Line 94. "……Among the several possible ways…..", please add some information on these ways.
- Lines 97-100, please add references.
- Page 3, line 104; “…and have S = T/P satellites”; what “S” is refers to?
- Please, enhance the artwork of Figure 1.
- Page 3; subtitle “Orbital elements” have no references, please give references for its first paragraph and equations 1 and 2. Also, give references of equations 3 and 4.
- Page 3, line 130; “in Figure 2 (right)”; where is figure 2? It is not included in the manuscript.
- Page 5, subtitle “Assessment of revisit rate and revisit time”; explain the method and equation used to calculate revisit rate and revisit time in details.
- Page 6, line 234; “…power system, energy generation must be equal to or greater…to the end of paragraph”; it is not understandable, what the relation of power to the present topic of manuscript? It is not our problem.
- Please, enhance the output of Figure 4.
Results:
- Page 6, line 254; “The revisit rate, mean revisit time, and maximum revisit time were calculated from” explain in details how they are calculated.
- Page 6, line 258; “…nearly linear correlation between…”, plot this correlation in a diagram and add it as a figure.
- Page 7, line 269; “…the mean revisit rates (i.e., the largest), the mean revisit time (i.e., the shortest), and the maximum revisit time (i.e., the shortest) for each T.” not understandable, review this sentence.
- Page 10, line 343; “…ratio of observed damage extent to damage extent…” 1- do you mean the observed damage by SAR data? Or by field observation? 2- If it is the observed damage by SAR data, then give an example by comparing the actual damage extent to the observed damage extent on a SAR image.
Discussion:
- Page 11, line 397; “…match the flood area…” plot the flood area on fig. 8.
Conclusions:
- Page 12, line 458; “…which is shown in this paper…” replace “is” by “are”.
Figures:
- Increase the resolution of all figures, they cannot be seen or read.
- Figure 2 is absent, please add it.
- Figure 4; add coordinates, scale, and north direction. Also the locations of GCP are not clear, please plot them clearly. It is preferable to add some details on the map like names of cities or regions.
- Figure 5; instead of (left, middle, and right) number them as (a, b, and c)
- Figure 8; plot the flood area. Also, clarify that the observed flood area is represented by the red color in the caption or as a legend.

Round 2
Reviewer 1 Report
The authors indeed did some improvements in this version. However, I still think that it is not suitable for publishing in Remote Sensing.
1) In the abstract, “few studies have focused on this potential application and a required observation performance with respect to flood disaster monitoring has been unclear.” On the contrary, flood monitoring is one of the most common concerns in SAR applications. Especially, in the ICEYE program, it has been adopted as the primary environmental monitoring application (please considering https://www.iceye.com/solutions/flood-monitoring). On the website, some briefings have verified the effectiveness of short revisiting observation. If the authors just reveal a potential application of an existed SAR constellation, the innovation is not convincing for a communication paper.
2) Section 2.3 should be an important input condition in this paper, which may be the authors’ adept domain. However, the discussion is incomplete. I would prefer to get the quantitative requirements for the constellation in this section, which can also solve these comments below. If the orbit design and constellation design are not the key points in this paper, sections 2.1 and 2.2 may mislead readers.
3) If the constellation proposed in this paper is just focusing on flood monitoring, why not choosing a lower orbital inclination? Flooding is rare in polar regions, and lower orbital inclination can also improve revisiting interval.
4) 3-m resolution is necessary for flood monitoring? If lower resolution can also satisfy the needs, ScanSAR mode may realize a shorter revisit.
5) How many satellites are enough for flood monitoring (or how long of the revisit interval is reasonable)? Is there a clear quantitative standard?
6) The figures are still not clear enough.
Reviewer 4 Report
Dear Mr. Nikola Stojanovic,
Thank you for giving me the opportunity to review this MS.
It is clear that the authors put a great deal of work into getting this manuscript to its present form. Although, the English text is revised but there are several errors exist, e.g., line 491, please, modify "an" to "a" before "… SAR small-satellite constellation". The figures still need revision. Therefore, I suggest that the paper can be accepted after minor revision.
